# Photocatalytic defluoroalkylation and hydrodefluorination of trifluoromethyls using *o*-phosphinophenolate

Can Liu[1], Ni Shen[1] & Rui Shang [1,2✉]

Under visible light irradiation, *o*-phosphinophenolate functions as an easily accessible photoredox catalyst to activate trifluoromethyl groups in trifluoroacetamides, trifluoroacetates, and trifluoromethyl (hetero)arenes to deliver corresponding difluoromethyl radicals. It works in relay with a thiol hydrogen atom transfer (HAT) catalyst to enable selective defluoroalkylation and hydrodefluorination. The reaction allows for the facile synthesis of a broad scope of difluoromethylene-incorporated carbonyl and (hetero)aromatic compounds, which are valuable fluorinated intermediates of interest in the pharmaceutical industry. The ortho-diphenylphosphino substituent, which is believed to facilitate photoinduced electron transfer, plays an essential role in the redox reactivity of phenolate. In addition to trifluoromethyl groups, pentafluoroethyl groups could also be selectively defluoroalkylated.

[1] Department of Chemistry, University of Science and Technology of China, Hefei 230026, China. [2] Department of Chemistry, The University of Tokyo, Tokyo 113-0033, Japan. ✉email: rui@chem.s.u-tokyo.ac.jp

Photoredox catalysis has demonstrated its strong reducing power to cleave unactivated chemical bonds via photo-induced electron transfer[1–5]. Among the various photocatalysis-enabled methods of activating inert bonds, the direct selective C–F activation[6–11] of trifluoromethyl groups to deliver corresponding difluoromethyl radicals is an ideal transformation for the synthesis of difluoromethylene-incorporated compounds, which are valuable fluorinated intermediates in the pharmaceutical industry[12–14]. The low cost and ready availability of trifluoroacetamide, acetate, and a variety of trifluoromethylated (hetero)aromatics make this transformation appealing. Ingenious photocatalytic methods for selective hydrodefluorination and defluoroalkylation of trifluoromethyl(hetero)arenes have been developed by Jui[15,16], König[17], and Gouverneur[18]. While the photocatalysts prevalently used in these transformations are precious metal-based polypyridyl complexes and π-conjugated organic dyes, we conceived that anionic phenolate[19–23], which has strong reductive potential in its excited state, may work as a suitable catalyst for selective C–F functionalization of trifluoromethyls over a broad scope of substrates. Although simple phenolate failed as a catalyst, probably because of its insufficient excited state lifetime and the poor stability of phenoxy radicals[19,21], which limited catalyst turnover, we were inspired by the effect of triphenylphosphine that we previously observed in photocatalytic decarboxylative couplings[24–27] to facilitate photo-induced electron transfer, and we hypothesized that an ortho-phosphino group easily installed onto phenolate may overcome these problems. We present our catalyst design in Fig. 1a. The installation of ortho-PPh₂ has three benefits. First, the ortho-PPh₂ substituent redshifts the absorption of the ground state anion (PO⁻) from the ultraviolet to visible light range. Second, phosphine exerts a heavy atom effect[28,29] to facilitate intersystem crossing to access the triplet state and extends its lifetime for efficient photoelectron transfer with substrates. Third, the interaction of phosphine with oxygen radicals is expected not only to facilitate efficient photoelectron transfer in the anionic excited state (*PO⁻) but also to stabilize the radical double state (PO•).

We show herein the application of o-phosphinophenolate for photocatalytic C–F activation of a wide range of trifluoromethyl groups in trifluoroacetamides, trifluoroacetates, and trifluoromethyl (hetero) arenes to deliver corresponding difluoromethyl radicals (Fig. 1b). In this work, we target two transformations, defluoroalkylation[15–17,30,31] with alkenes and hydrodefluorination[18,32,33], to prepare valuable carbonyl and aromatic compounds with incorporated difluoromethylene motifs.

## Results and discussion

**Studies of properties of PO catalysts and reaction parameters.** Our mechanistic hypothesis depicted in Fig. 1c was inspired by previous examples of photoredox/HAT synergistic catalysis[15,16,18]. The photoexcited PO⁻ catalyst (*PO⁻) reduces the trifluoromethyl substrate (I) to deliver difluoromethyl radicals (II). II can be reduced by a hydrogen donor to produce a hydrodefluorination product or be intercepted by an alkene to generate a new alkyl radical (III). The alkyl radical (III) can be reduced by a polarity-reversal thiol HAT catalyst[34] to produce a defluoroalkylation product. The thiol catalyst regenerates through hydrogen abstraction from formate (BDE of formate C–H: 88 kcal/mol) to deliver $CO_2^{•-}$, which is a strong reductant ($E_{1/2}$ $CO_2/CO_2^{•-}$ = −2.2 V vs. SCE), to reduce PO• and complete the PO⁻ redox cycle. Guided by the hypothesis, we first focused on the selective defluoroalkylation of trifluoroacetamide with alkenes, a transformation that has not been successfully developed using photocatalysis. We discovered that a catalytic amount of PO in combination with 1-adamantanethiol (1-AdSH) as the HAT catalyst in

the presence of formate catalyzed monoselective defluoroalkylation of N-phenyltrifluoroacetamide with alkenes under irradiation with a 427 nm LED (Kessil®, emission FWHM of ~20 nm)[35]. A similar transformation was reported by Houk and Wang et al. under thermal conditions using a stoichiometric amount of DMAP-BH₃ via a spin-center shift strategy[36]. Figure 2 shows the key reaction parameters. Under optimal conditions (Fig. 2a), defluoroalkylation product (3) was obtained in 90% yield determined by ¹H-NMR, along with the generation of 4 in 5 % yield. The reaction mixture appeared as a light yellow homogeneous solution, suggesting its promising applications in flow photosynthesis[37,38]. UV–Vis absorption spectra of catalysts, substrates, and reaction mixtures were measured to determine the light absorbing species (Fig. 2b). Both substrates (1 and 2) absorb light only in the ultraviolet range (< 325 nm). PO1 in its phenol form had an absorption onset at ~370 nm (purple line in Fig. 2b). Upon mixing with Cs₂CO₃, the deprotonated anion of PO1 (PO1⁻) exhibited redshifted absorption into the visible light range with an absorption onset at ~425 nm. Comparison of the absorption spectra of PO1⁻ and tert-butylphenolate showed that the ortho-PPh₂ substituent significantly redshifted the absorption curve by ~70 nm (see Supplementary Information for details). PO1⁻ absorbs visible light with an onset at ~425 nm and shows an emission maximum at 540 nm. The reduction potential of excited PO1⁻ (*PO1⁻) is estimated to be −2.89 V vs. SCE, a value sufficient to reduce a broad scope of trifluoromethyl aromatic and carbonyl compounds

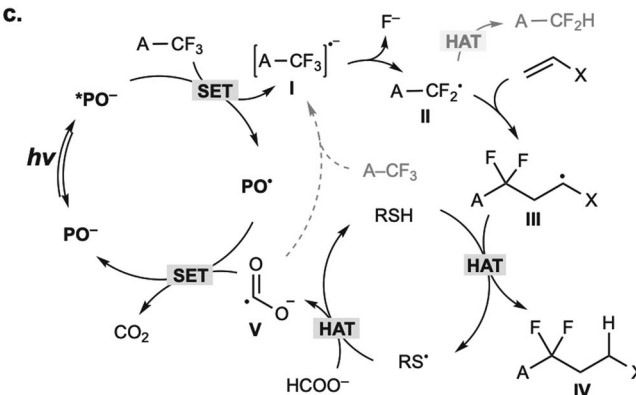

**Fig. 1 o-Phosphinophenolate as photocatalyst for C–F functionalization of –CF₃. a** Design of a system that uses o-phosphinophenolate (PO⁻) as a photoredox catalyst. **b** Generation of –CF₂• radicals from –CF₃ groups using PO⁻. **c** Hypothesized photocatalytic cycles for defluoroalkylation and hydrodefluorination. hv light, SET single electron transfer, HAT hydrogen atom transfer.

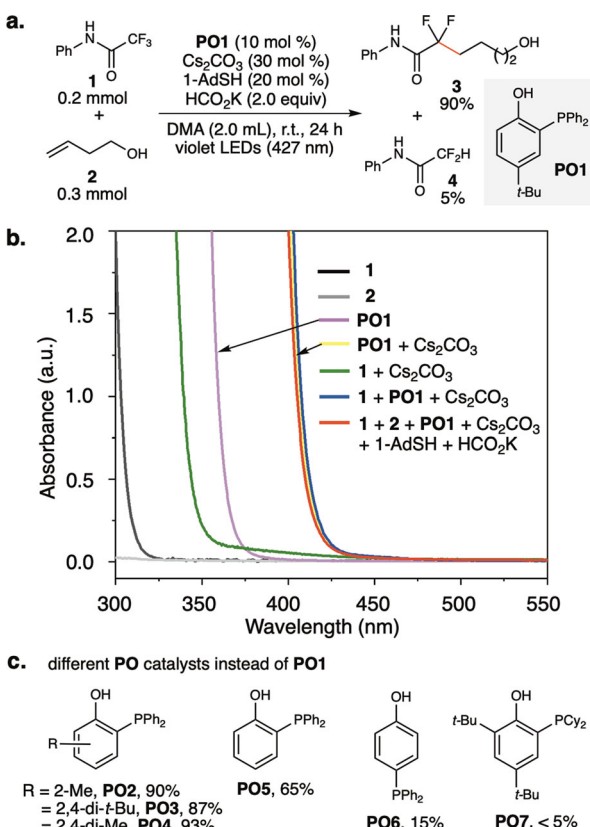

**Fig. 2 Defluoroalkylation of trifluoroacetamide with an alkene. a** Optimized reaction condition. **b** UV–Vis absorption spectra of catalyst, substrates, and reaction mixtures. The concentrations of substances measured were identical to those under optimized reaction conditions in Fig. 2a. Arbitrary units (a.u.) **c** PO catalysts other than **PO1**.

**Table 1 Key reaction-controlling parameters.**

| Entry | Variations from standard condition | Yield$^a$ (%) |
|---|---|---|
| 1 | CySH instead of 1-AdSH | 75 |
| 2 | 1-AdSH (10 mol %) | 72 |
| 3 | **PO1** (2 mol %) | 52 |
| 4 | **2** (0.4 mmol) | 96 |
| 5 | Et$_3$SiH instead of HCO$_2$K | 30 |
| 6 | 440 nm instead of 427 nm | 70 |
| 7 | 456 nm instead of 427 nm | trace |
| 8 | 467 nm instead of 427 nm | trace |
| 9 | PPh$_3$ instead of **PO1** | trace |
| 10 | 4-$t$-Bu-C$_6$H$_4$OH instead of **PO1** | 0 |
| 11 | 4-$t$-Bu-C$_6$H$_4$OH+PPh$_3$ (1:1, 10 mol %) | 10 |
| 12 | *w/o* **PO1** | 0 |
| 13 | *w/o* 1-AdSH | trace |
| 14 | *w/o* HCO$_2$K | 18 |
| 15 | *w/o* light | 0 |
| 16 | under air | trace |

*w/o* without.
$^a$Yields measured by $^1$H-NMR using diphenylmethane as an internal standard.

(PhCF$_3$, E$^{red}_{1/2}$ = −2.50 V vs. SCE in DMF[18]; CF$_3$COOEt, E$^{red}_{p/2}$ = −2.40 V vs. SCE) (see Supplementary Information S41, for details). Anions generated by the deprotonation of amide **1** maximally absorb light until 350 nm. In the presence of **1**, the absorption spectrum of **PO1**$^-$ did not show a bathochromic shift, showing that an electron-donor-acceptor complex[39–41] between **1** and **PO1**$^-$ was not formed (blue line). The absorption curve of the reaction mixture (red line) was identical to that of **PO1**$^-$, indicating that **PO1**$^-$ was the light absorbing species in the reaction mixture. Figure 1c shows the PO catalysts with different structures. 2-Methyl (**PO2**)-, 2,4-di-*tert*-butyl (**PO3**)-, and 2,4-dimethyl (**PO4**)-substituted *o*-phosphinophenols all showed performances comparable to that of **PO1**. *o*-Phosphinophenol (**PO5**) without a *p*-substituent showed reduced catalytic efficacy. *p*-Phosphinophenol (**PO6**) gave **3** in only 15% yield, which suggests the essential role of intramolecular P–O interactions in catalytic efficacy. A bulky PO catalyst with a dicyclohexylphosphine substituent (**PO7**) was ineffective. Regarding to stability, **PO1** is bench-stable white powder and can be stored under ambient air for months without apparent decomposition and oxidation.

Table 1 summaries key reaction parameters. Using cyclohexanethiol as the HAT catalyst reduced the yield (entry 1, Table 1). Using 10 mol % 1-AdSH resulted in decreased yield (entry 2). **PO1** (2 mol %) still catalyzed the reaction in 52% yield (entry 3). Increasing the amount of alkene to 2.0 equivalents suppressed hydrodefluorination and increased the defluoroalkylation yield to 96%. Replacing formate with triethylsilane gave **2** in 30% yield (entry 5), indicating that formate is not essential for C–F activation. LEDs with an emission peak at 440 nm (emission range from 415 nm to 470 nm) also promoted the reaction

(entry 6), but LEDs with emission peaks at 456 nm and 467 nm, which did not overlap with the absorption of **PO1**$^-$, were ineffective (entries 7 and 8). PPh$_3$ and 4-tert-butylphenol used alone were both ineffective (entries 9 and 10), while 10% of **3** was generated by using a mixture of them (entry 11), suggesting certain role of P–O interactions in efficient photoelectron transfer. Control experiments showed that **PO1**, 1-AdSH, and light were all essential parameters (entries 12, 13, 15). In the absence of formate salt, product **3** was detected in 18% yield (entry 14), suggesting the role of formate in catalyst turnover (ref. Fig. 1c). The cation moiety of formate salt affects not only solubility but also reactivity, because the alkali metal cations act as counter cations of both formate and generated fluoride salt, that may affect the rates of HAT and defluorination. Hence, different formate salts (Li, Na, K, Cs) were tested (see Supplementary Table 1 in SI page 6 for details). The quantum yield of **3** was estimated to be 4.4 according to the literature[42,43], which suggested that CO$_2$•$^-$ generated after HAT may activate –CF$_3$ substrates (e.g., reduction potential of 1,3-bistrifluoro-methylbenzene, E$^{red}_{1/2}$ = −2.07 V vs. SCE; reduction potential of **1**, E$^{red}_{p/2}$ = −2.11 V vs. SCE) in relay with the thiol HAT catalyst to deliver **3** (pale dashed arrow in Fig. 1c)[44,45]. Exposure to air completely killed the catalytic reactivity (entry 16) resulting recovery of starting materials, as air can quench excited triplet state of **PO1**$^-$ and oxidize thiol.

**Scope of the reactions**. Figure 3 illustrates the scope of the defluoroalkylation of trifluoroacetamides. The reaction can be easily scaled up to the gram scale under batch conditions using Kessil LEDs and a Schlenk flask (**3**). For trifluoroacetamides possessing electron-neutral and electron-rich *N*-aryl substituents, monodefluoroalkylation products accompanied by a small amount of hydrodefluorination byproduct (5–10%) were obtained (**5, 6, 7**). For trifluoroacetamides with strong electron-deficient *N*-aryl substituents, products of didefluoroalkylation were obtained as major products (**8, 9**), and monodefluoroalkylation products were observed only in trace amounts (<3%). The recovered amide staring materials accounted for moderate yields (**8, 9**). For *N*-3-

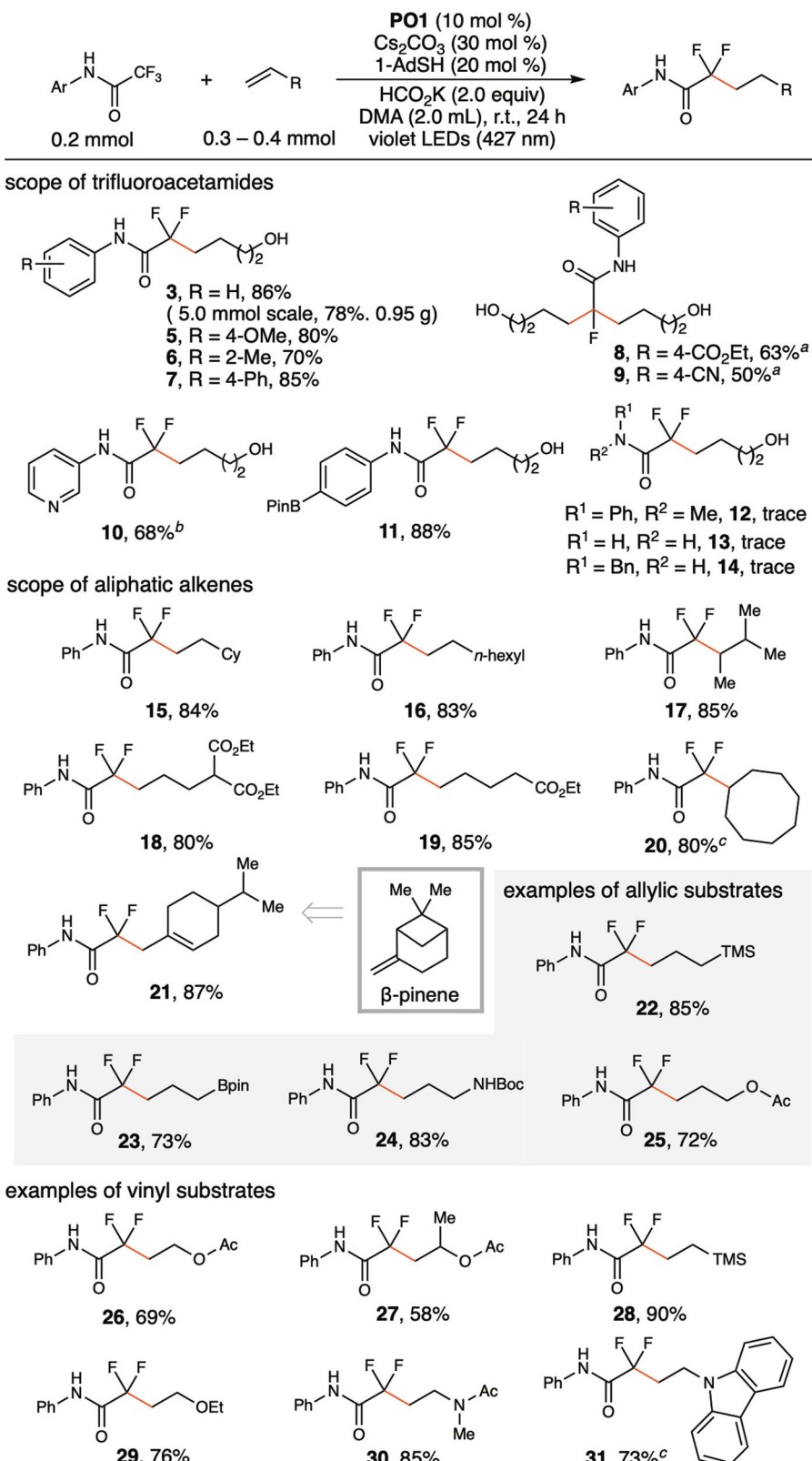

**Fig. 3 Scope of defluoroalkylation of trifluoroacetamides.** Yields of isolated products. [a]alkene (0.5 mmol), HCO$_2$K (0.6 mmol). [b]**PO3** (10 mol %) used instead of **PO1**. [c]alkene (0.4 mmol).

pyridyl trifluoroacetamide (**10**), a monodefluoroalkylation product was obtained in 68% yield.

An *N*-aryl substituent is essential for defluoroalkylation (**13**, **14**), and the reactivity is not applicable to tertiary amides (**12**).

Measuring reduction potentials of different trifluoroacetamides by cyclic voltammetry revealed that *N*-phenyltrifluoroacetamide (**1**) is thermodynamically easier to reduce ($E^{red}_{p/2}$ = –2.11 V vs. SCE) than *N*-benzyltrifluoroacetamide ($E^{red}_{p/2}$ = –2.56 V vs.

SCE) and primary trifluoroacetamide ($E^{red}_{p/2} = -2.49$ V vs. SCE). We also rationalize that *N*-aryl substituent stabilizes amide radical anion through charge delocalization, thus suppresses back electron transfer with **PO•** to facilitate subsequent cleavage of C–F bond. The reaction tolerates a broad scope of aliphatic alkenes (**15, 16, 18, 19**), including sterically hindered internal alkenes (**17**) and cyclic alkenes (**20**). *β*-Pinene gave ring-opened product (**21**) in high yield. Allylic-type substrates were also amenable, as allylic silane (**22**), allylic boronate (**23**), allylic amine (**24**), and allylic acetate (**25**) all reacted well. The amenable alkenes also include vinyl-type substrates. Vinyl acetates (**26, 27**) are suitable substrates. Hydrolysis of defluoroalkylation products with vinyl acetates can provide fluorinated and methylated *γ*-hydroxybutyrate (**26, 27**), which is an active component of XYREM, a drug approved by the FDA to treat symptoms of narcolepsy[46]. Vinyl silane (**28**), vinyl ether (**29**), *N*-vinyl amide (**30**), and vinyl carbazole (**31**) all reacted effectively. However, styrene- and acrylate-type substrates were unsuitable, probably because of the facile addition of these types of alkenes with $CO_2$•[−] under the reaction conditions[47,48]. In addition to trifluoroacetamides, trifluoroacetates also reacted in high yield to provide α,α-difluorinated aliphatic carboxylate esters (Fig. 4, **32–35**).

In the absence of an alkene, hydrodefluorination products were generated (Fig. 5). Thiol as a polarity reversal catalyst[34] is essential for a high yield of hydrodefluorination. Cesium formate used in 1.2 equivalents is critical to ensure high monoselectivity (**4, 36–39**). The *N*-phenyltrifluoroacetamide with a *para*-ester substituent underwent thorough hydrofluorination to generate acetamide (**40**). The same reaction conditions are also applicable to the selective hydrodefluorination and defluoroalkylation of pentafluoropropionamide (Eqs. (1) and (2) in Fig. 6), suggesting the further application of **PO** catalysts in the selective C–F functionalization of polyfluorinated compounds (**41, 43**)[49–51].

In Fig. 7, the application of **PO1** as a catalyst for defluoroalkylation of trifluoromethyl arenes and heteroarenes is demonstrated. Lithium formate was found to be the hydrogen donor of choice, and DMSO was found to be the preferred solvent. The monoselective defluoroalkylation reaction is applicable to a broad scope of trifluoromethyl arenes, including both *di*-CF₃ substituted arenes and *mono*-CF₃ substituted arenes. Potassium *tert*-butoxide was found to be a better base for *di*-CF₃ arenes than cesium carbonate because it resulted in high selectivity (**44, 45, 46, 48, 49**). Although trifluoromethylbenzene was an unsuitable substrate (<15% conversion), *m*-methoxy-substituted trifluoromethylbenzene reacted in 50% yield (**50**). *m*-CF₃-substituted phenylboronate reacted in 78% to give fluorinated building blocks useful in cross-coupling reactions (**51**). Pyridine derivatives bearing –CF₃ at both the 2- and 3-positions reacted smoothly, generating valuable α,α-difluoroalkylated pyridines (**52, 53**). For alkene scope, in addition to vinyl silane (**54**) and aliphatic alkene (**57**), acetals of acrylaldehyde (**55**), vinyl ethylene carbonate (**56**), *N*-vinylpyrrolidinone (**58**), and 2-vinyloxytetrahydropyran (**59**) all reacted smoothly. An FDA-approved antipsychotic drug for schizophrenia, trifluoperazine (Stelazine®), was selectively defluoroalkylated (**60**), showcasing the practical utility for late-stage functionalization of CF₃-containing drugs. Monoselective hydrodefluorination of trifluoromethyl arene could also be achieved in the absence of alkene (**61**, Fig. 8).

In summary, *o*-phosphinophenolates were developed as efficient photocatalysts for the selective C–F activation of a wide range of trifluoromethyl groups in trifluoroacetamides, trifluoroacetates, and trifluoromethyl (hetero) arenes to deliver corresponding difluoromethyl radicals for defluoroalkylation with alkenes or hydrodefluorination to prepare valuable difluoromethylene-incorporated carbonyl and aromatic compounds. Similar reactivity is also

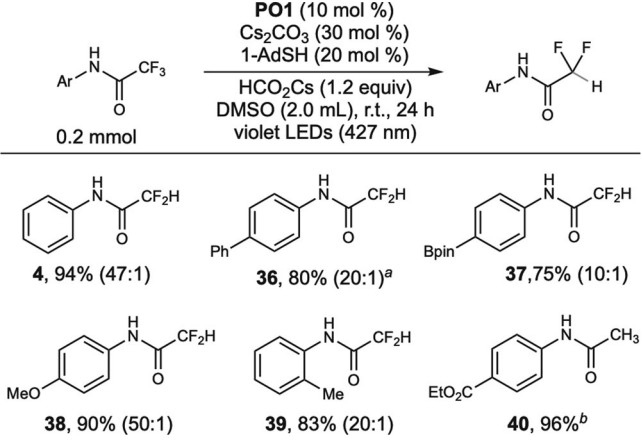

**Fig. 4 Examples of defluoroalkylation of trifluoroacetates.** Yields of isolated products.

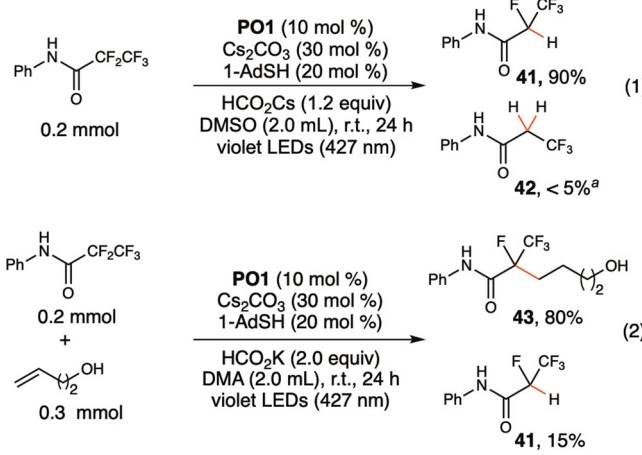

**Fig. 5 Scope of hydrodefluorination of trifluoroacetamides.** Yields of isolated products. *[a]*PO5 (10 mol %), HCO₂Cs (0.4 mmol). *[b]*HCO₂Cs (0.8 mmol). The ratios of monohydrodefluorination and dihydrodefluorination are shown in parentheses.

applicable for selective functionalization of pentafluoroethyl groups. In these reactions, *o*-phosphinophenolate works synergistically with an alkyl thiol HAT catalyst[52]. This work offers practical methods of synthesizing valuable geminal difluoro-substituted carbonyl and

**Fig. 7 Scope of defluoroalkylation of trifluoromethyl(hetero)arenes.** Yields of isolated products. [a]KO[t]Bu (30 mol %). [b]**PO3** (10 mol %) instead of **PO1**, alkene (0.6 mmol).

aromatic compounds and demonstrates a new design strategy for developing photoredox catalysts at a low cost (We list prices of typical photoredox catalysts and prices of materials for **PO** synthesis to justify this statement. From Aldrich® (Japan), 4CzIPN, 250 mg, 1282 USD; (Ir[dF(CF₃)ppy]₂(dtbpy))PF₆, 1 g, 1500 USD; (2-hydroxyphenyl)diphenylphosphine (**PO5**), 1 g, 135 USD. Materials for synthesis of **PO1**: 2-bromo-4-*tert*-butylphenol, 5 g, 62 USD; chlorodiphenylphosphine, 25 g, 60 USD).

## Methods

**Materials**. For the optimization of reaction conditions, see Supplementary Tables 1 and 2, For the experimental procedures and analytic data of compounds synthesized, see Supplementary Methods. For NMR spectra of compounds in this manuscript, see Supplementary Figures 13–179.

**General procedure for defluoroalkylation of trifluoroacetamides**. Tri-fluoroacetamide (1.0 equiv, 0.2 mmol), alkene (1.5 equiv., 0.3 mmol), **PO1** (10 mol %), Cs₂CO₃ (30 mol %), 1-adamananethiol (20 mol %), HCO₂K (2.0 equiv., 0.4 mmol)

**Fig. 8 Monoselective hydrodefluorination of trifluoromethyl arene.** Yield of isolated product. The ratio of monohydrodefluorination and dihydrodefluorination measured by $^{19}$F-NMR is shown in parentheses.

were placed in a transparent Schlenk tube equipped with a stirring bar. The tube was evacuated and filled with argon (three times). To the mixture, anhydrous DMA (2 mL) were added via a gastight syringe under argon atmosphere. The reaction mixture was stirred under irradiation with violet LEDs (Kessil® 427 nm) in a HepatoChem photoreactor at room temperature for 24 h. The mixture was quenched with brine and extracted with ethyl acetate (3 × 10 mL). The organic layers were combined and concentrated under vacuo. The product was purified by flash column chromatography on silica gel.

## Data availability

The authors declare that all the data supporting the findings of this study are available within the paper and its supplementary information files, or from the corresponding author upon request.

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

## Acknowledgements
This work was financially supported by the National Natural Science Foundation of China and USTC (GG2065010002 and KY2060000119).

## Author contributions
R.S. conceived the idea and wrote the manuscript. C.L. performed the experiments. C.L. and N.S. analyzed the data and participated in preparation of the manuscript.

## Competing interests
The authors declare no competing interests.
