## [Peer Review File · Nature Communications]

Reviewers' Comments:

Reviewer #1:

Remarks to the Author:

This manuscript describes the defluoroalkylation and hydrodefluorination of the trifluoromethyl group under violet LEDs using o-phosphinophenolate as the photocatalyst and thiol HAT as a catalyst in yields ranging from 50 to 96% yield. Although this reported method (trifluoroacetamides) is similar to a previously reported transformation by Houk and Wang et al. (10.1126/science.abg0781) using a stoichiometric amount of DMAP-BH₃ with catalytic amounts of THBH as a radical initiator, the key difference is the replacement of the mediator with catalytic amounts of o-phosphinophenolate as the photocatalyst and light. The same can be said for the scope of trifluoromethyl(hetero)arenes where Jiu et al. (10.1021/jacs.9b06004) disclosed a similar defluorofunctionalization with a highly reducing photocatalyst and thiol HAT.

In my opinion, the novelty of this work lies in the use of o-phosphinophenolate as a photocatalyst as well as the milder reaction conditions (e.g. room temperature) compared to previously reported conditions (50-120 °C). The reaction shows a broad substrate scope, high functional group tolerance, and can be scaled up to a gram scale. This approach allows the rapid construction of fluorine-containing products from a variety of trifluoromethylated precursors. Overall, if the following revisions can be properly addressed, I would recommend for publication in a top journal such as Nat. Commun.

- The authors reported a quantum yield of 4.4 which indicates the possibility of a radical chain reaction. In Figure 1c in the manuscript the hypothesized photocatalytic cycle is reported as a closed catalytic cycle. Is it possible that abstraction of a hydrogen atom from formate to generate CO₂•⁻ (V) (-2.2 V vs. SCE) which can directly reduce A-CF₃ thereby propagating the radical chain process? Would be the main pathway instead? Reference of reduction potential of the reactants or CV should be mentioned and discussed.

-The authors mentioned "the reduction potential of excited PO1- (*PO1-) is estimated to be -2.89 V vs. SCE, a value sufficient to reduce a broad scope of trifluoromethyl aromatic and carbonyl compounds (see Supporting Information S39, for details)." The statement "sufficient to reduce a broad scope of trifluoromethyl aromatic and carbonyl compounds" should include a reference and/or CV of the respective substrate in question to support this statement.

-Stern-Volmer quenching study is incomplete, only quenching study of 1 was reported. Evaluation of 2 and thiol HAT catalyst should also be done and compared with 1 and discussed.

-It would be valuable to discuss the stability of PO1 and whether this catalyst is stable or prone to oxidation under air. Mass balance for the reaction in Table 1 entry 16 (under air) would be helpful in this regard.

-The potential of late-stage functionalization under this catalytic system should be evaluated and demonstrated.

-All ¹⁹F NMR spectra of the purified final products should also include a zoom of the region containing the signals of interest.

-There is a discrepancy between ¹⁹F NMR of 27: S75 shows -102.9 to -105.91 while in S20 the ¹⁹F NMR of 27 is reported as δ -90.5 – -120.1. Please clarify.

Minor comments:

-The term purple LEDs in the manuscript and supporting information should be changed to violet LEDs. There is no light wavelength that corresponds to purple (the color is from the combinations of red and blue) whereas violet is a spectral color (e.g. referring to the color of different single wavelengths of light)

- Authors should clarify which nuclei for the NMR was this yield was determined by in "...product (3)

was obtained in 90% yield by NMR, along with the generation of 4 in 5 % yield.”

-There is no label for the scheme shown in under Fig. 5 and 6.

-In the supporting information compounds 5 and 36 are reported as in DMSO, does the author mean d6-DMSO?

- Cost analysis vs metal and organic photocatalyst should be done to support this statement of “This work offers practical methods of synthesizing valuable geminal difluoro-substituted carbonyl and aromatic compounds and demonstrates a new design strategy for developing photoredox catalysts at a low cost.”

Reviewer #2:

Remarks to the Author:

The authors report a new type of photoredox catalyst and its utilization for defluoroalkylation/hydration of trifluoromethyl compounds. They have synthesized phosphinophenol as a very reactive photoredox catalyst especially for reducing process. This property realized the amides or esters bearing CF₃ group which have not been applied to this type of reaction before.

The design of the catalyst is quite interesting. It can be a new trend of the photoredox catalyst because organic moiety would be tunable for various ways.

Variety of substrate scope and mechanistic discussion are reasonable.

The cationic part of the HCO₂M seems to be important because each system has suitable metal ion. The author should describe the matter.

The reviewer believe the manuscript is suitable for publication in Nature Communications after revision.

Reviewer #3:

Remarks to the Author:

The introduction of gem-difluoromethylene group into organic compounds has received a great attention and big progress has been made recently. Among these progress, the selective C-F functionalization of trifluoromethylated compounds through photoredox catalysis has received increasing attention very recently. In the most of the published work, the precious metal-based complexes and trifluoromethylated (hetero)arenes were often used as the catalyst and fluorinated material respectively. In this manuscript, the authors described the photocatalysis activation the trifluoromethyl groups of trifluoroacetamides, trifluoroacetates and trifluoromethylated (hetero)arenes using o-Phosphinophenolate as the catalyst. This protocol provides a practical and efficient route to gem-difluoromethylene group containing organic compounds. Although the very useful synthetic method was developed on the basis of the author's published work (Ref. 24-27), the design and using of o-phosphinophenolate was very novel. The authors have made an important contribution to photoredox catalysis and fluorine chemistry. Of course, this manuscript was recommended for publication in Nature Communication when the following comments were addressed.

1) Please explain in detail why the N-aryl substituent of trifluoroacetamides was essential for the defluoroalkylation (Fig 3. substrates 12-14).

2) Regarding to Table 1 and Fig. 1c, From the proposed reaction mechanism, the reaction should proceed smoothly in the presence of HCO₂K without 1-AdSH, as the intermediate III can be reduced by HCO₂K for the formation of product IV (Fig. 1c). Please explain why 1AdSH was essential (Table 1, entry 13).

Reviewer #1 (Remarks to the Author):

This manuscript describes the defluoroalkylation and hydrodefluorination of the trifluoromethyl group under violet LEDs using o-phosphinophenolate as the photocatalyst and thiol HAT as a catalyst in yields ranging from 50 to 96% yield. Although this reported method (trifluoroacetamides) is similar to a previously reported transformation by Houk and Wang et al. (10.1126/science.abg0781) using a stoichiometric amount of DMAP-BH₃ with catalytic amounts of THBH as a radical initiator, the key difference is the replacement of the mediator with catalytic amounts of o-phosphinophenolate as the photocatalyst and light. The same can be said for the scope of trifluoromethyl(hetero)arenes where Jiu et al. (10.1021/jacs.9b06004) disclosed a similar defluorofunctionalization with a highly reducing photocatalyst and thiol HAT.

In my optional, the novelty of this work lies in the use o-phosphinophenolate as a photocatalyst as well as the milder reaction conditions (e.g room temperature) compared to previously reported conditions (50-120 °C). The reaction shows a broad substrate scope, high functional group tolerance, and can be scaled up to a gram scale. This approach allows the rapid construction of fluorine-containing products from a variety of trifluoromethylated precursors. Overall, if the following revisions can be properly addressed, I would recommend for publication in a top journal such as Nat. Commun.

Response: We thank this reviewer for the positive evaluation and his/her support to publish our manuscript in Nat. Commun..

- The authors reported a quantum yield of 4.4 which indicates the possibility of a radical chain reaction. In Figure 1c in the manuscript the hypothesized photocatalytic cycle is reported as a closed catalytic cycle. Is it possible that abstraction of a hydrogen atom from formate to generate CO₂•⁻ (V) (-2.2 V vs. SCE) which can directly reduce A-CF₃ thereby propagating the radical chain process? Would be the main pathway instead? Reference of reduction potential of the reactants or CV should be mentioned and discussed.

Response: We thank this reviewer for this insightful comment. Indeed, based on the measured quantum yield of 4.4, we also conceived that a radical chain process involving reduction of A-CF₃ by CO₂•⁻ concurrently exists. As shown in Figure 1c, the dashed arrow indicates this process. We commented this possibility in our manuscript as *“The quantum yield of **3** was estimated to be 4.4 according to the literature,⁴²⁻⁴³ which suggested that CO₂•⁻ generated after HAT may activate -CF₃ substrates in relay with the thiol HAT catalyst to deliver **3** (pale dashed arrow in Figure 1c).⁴⁴⁻⁴⁵”*. However, we do not consider this chain process as a main process, because light/dark experiments showed that product formation stopped in the absence of light, which indicates that light excitation for chain initiation is not sufficient for the reaction to proceed. Judged from reduction potential (e.g. for 1,3-bistrifluoromethylbenzene, E_{1/2} = -2.07 V vs SCE), CO₂•⁻ is feasible to reduce Ar-CF₃. We also performed CV measurement of trifluoroacetamide **1**. Reduction potential of **1** is estimated to be E_{1/2} = -2.11 V vs SCE, a value indicating a reduction by CO₂•⁻ feasible. We added the values of reduction potentials in the revised manuscript to support.

-The authors mentioned “the reduction potential of excited PO1– (*PO1–) is estimated to be -2.89 V vs. SCE, a value sufficient to reduce a broad scope of trifluoromethyl aromatic and carbonyl compounds (see Supporting Information S39, for details).” The statement “sufficient to reduce a broad scope of trifluoromethyl aromatic and carbonyl compounds” should include a reference and/or CV of the respective substrate in question to support this statement.

Response: We thank this reviewer to point out this issue. We added redox potential of $-\text{CF}_3$ compound and cited a corresponding reference to support this statement. Redox potential of PhCF_3 is estimated to be -2.50 V vs SCE in DMF (reported by Gouverneur et al. *J. Am. Chem. Soc.* **2020**, *142*, 9181), which is added as a reference.

-Stern-Volmer quenching study is incomplete, only quenching study of **1** was reported. Evaluation of **2** and thiol HAT catalyst should also be done and compared with **1** and discussed.

Response: According to this suggestion, we performed additional Stern-Volmer quenching studies. Alkene (**2**) and AdSH showed no apparent fluorescence quenching effect. This information is added to the revised Supplementary Information.

-It would be valuable to discuss the stability of PO1 and whether this catalyst is stable or prone to oxidation under air. Mass balance for the reaction in Table 1 entry 16 (under air) would be helpful in this regard.

Response: PO1 is bench-stable white powder and can be stored under air for months without apparent decomposition and oxidation. We observed only recovery of both starting materials in Table 1 entry 16. Photoexcited PO1 can be quenched by molecular oxygen. We include this information in the revised manuscript.

-The potential of late-stage functionalization under this catalytic system should be evaluated and demonstrated.

Response: We thank this reviewer for suggestion to test late-stage functionalization. We added one example of late-stage selective C-F functionalization of a medicinal compound, trifluoperazine (Stelazine[®]), in our revised manuscript (Figure 6) to showcase the practical utility of our catalytic system.

-All ^{19}F NMR spectra of the purified final products should also include a zoom of the region containing the signals interest.

Response: Revised accordingly.

-There is a discrepancy between ^{19}F NMR of **27**: S75 shows -102.9 to -105.91 while in S20 the ^{19}F NMR of **27** is reported as δ -90.5 – -120.1. Please clarify.

Response: We corrected this mistake. ^{19}F -NMR of **27** should be -102.9 to -105.91.

Minor comments:

-The term purple LEDs in the manuscript and supporting information should be changed to violet LEDs. There is no light wavelength that corresponds to purple (the color is from the combinations of red and blue) whereas violet is a spectral color (e.g referring to the color of different single wavelengths of light)

Response: We appreciate this reviewer pointing out this issue, which we are unaware of. We revised all the “purple LEDs” in our manuscript to “violet LEDs (427 nm)”.

- Authors should clarify which nuclei for the NMR was this yield was determine by in “..product (3) was obtained in 90% yield by NMR, along with the generation of 4 in 5 % yield.”

Response: We clarified in the revised manuscript that we used ^1H -NMR.

-There is no label for the scheme shown in under Fig. 5 and 6.

Response: We added scheme titles for these two schemes in the revised manuscript.

-In the supporting information compounds 5 and 36 are reported as in DMSO, does the author mean d_6 -DMSO?

Response: Yes, it is d_6 -DMSO and we revised accordingly.

- Cost analysis vs metal and organic photocatalyst should be done to support this statement of “This work offers practical methods of synthesizing valuable geminal difluoro-substituted carbonyl and aromatic compounds and demonstrates a new design strategy for developing photoredox catalysts at a low cost.”

Response: We list prices of typical photoredox catalysts and of materials for **PO** synthesis to justify this statement in reference 53.

From Aldrich® (Japan), 4CzIPN, 250 mg, 1282 USD; ($\text{Ir}[\text{dF}(\text{CF}_3)\text{ppy}]_2(\text{dtbpy})\text{PF}_6$), 1 g, 1500 USD; (2-hydroxyphenyl)diphenylphosphine (**PO5**), 1g, 135 USD.

Materials for synthesis of **PO1**: 2-bromo-4-*tert*-butylphenol, 5g, 62 USD; chlorodiphenylphosphine, 25g, 60 USD.

We appreciate this reviewer to spend his/her precious time to review our manuscript. The reviewer comments helped us to improve our work significantly.

Reviewer #2 (Remarks to the Author):

The authors report a new type of photoredox catalyst and its utilization for defluoroalkylation/hydration of trifluoromethyl compounds. They have synthesized phosphinophenol as a very reactive photoredox catalyst especially for reducing process. This property realized the amides or esters bearing CF₃ group which have not been applied to this type of reaction before.

The design of the catalyst is quite interesting. It can be a new trend of the photoredox catalyst because organic moiety would be tunable for various ways.

Variety of substrate scope and mechanistic discussion are reasonable.

Response: We thank this reviewer for these positive comments. Especially, we are very grateful to see the reviewer considers our catalyst design can be a new trend of photoredox catalyst.

The cationic part of the HCO₂M seems to be important because each system has suitable metal ion. The author should describe the matter.

Response: The cation moiety of formate salt affects not only solubility but also reactivity. As counter cation of both fluoride and formate, the alkali metal cation may affect the rates of defluorination and HAT. However, selection of HCOOM in our work was based on experimental trials evaluating product yields.

The reviewer believe the manuscript is suitable for publication in Nature Communications after revision.

Response: We thank this reviewer for supporting publishing our work in Nature Communications.

Reviewer #3 (Remarks to the Author):

The introduction of gem-difluoromethylene group into organic compounds has received a great attention and big progress has been made recently. Among these progress, the selective C-F functionalization of trifluoromethylated compounds through photoredox catalysis has received increasing attention very recently. In the most of the published work, the precious metal-based complexes and trifluoromethylated (hetero)arenes were often used as the catalyst and fluorinated material respectively. In this manuscript, the authors described the photocatalysis activation the trifluoromethyl groups of trifluoroacetamides, trifluoroacetates and trifluoromethylated (hetero)arenes using *o*-Phosphinophenolate as the catalyst. This protocol provides a practical and efficient route to gem-difluoromethylene group containing organic compounds. Although the very

useful synthetic method was developed on the basis of the author's published work (Ref. 24-27), the design and using of o-phosphinophenolate was very novel. The authors have made an important contribution to photoredox catalysis and fluorine chemistry. Of course, this manuscript was recommended for publication in Nature Communication when the following comments were addressed.

Response: We thank this reviewer for the very positive evaluation and supporting publication in Nature Communications. Especially, we appreciate the encouraging comments evaluating our catalyst design as “very novel”.

1) Please explain in detail why the N-aryl substituent of trifluoroacetamides was essential for the defluoroalkylation (Fig 3. substrates 12-14).

Response: We thank this very insightful comments.

Measuring reduction potentials of different trifluoroacetamides by cyclic voltammetry revealed that N-phenyl trifluoroacetamide is thermodynamically easier to reduce (reduction potential of **1**, $E_{p/2}^{\text{red}} = -2.11$ V vs SCE) than N-benzyl trifluoroacetamide ($E_{p/2}^{\text{red}} = -2.56$ V vs SCE) and primary trifluoroacetamide ($E_{p/2}^{\text{red}} = -2.49$ V vs SCE). We also consider that N-aryl substituent stabilizes amide radical anion generated after electron transfer through charge delocalization, thus suppresses back electron transfer, and facilitates subsequent cleavage of C-F bond.

We added this explanation in the revised manuscript.

2) Regarding to Table 1 and Fig. 1c, From the proposed reaction mechanism, the reaction should proceed smoothly in the presence of HCO₂K without 1-AdSH, as the intermediate III can be reduced by HCO₂K for the formation of product IV (Fig. 1c). Please explain why 1AdSH was essential (Table 1, entry 13).

Response: As shown by experimental result in Table 1, entry 13, thiol catalyst was essential. Thiol is used as polarity-reversal catalyst to facilitate HAT process. Although intermediate III may be reduced by HCO₂K to deliver product IV, the polarity matching issue in the HAT transition state causes a very slow rate making this process not productive. Thiol as polarity-reversal catalyst is known to solve this problem (see, reference 34 cited, “Polarity-reversal catalysis of hydrogen-atom abstraction reactions: concepts and applications in organic chemistry”. *Chem. Soc. Rev.* 28, 25–35 (1999)). We commented the effect of AdSH as polarity-reversal catalyst and cited this review article in our manuscript.

Reviewers' Comments:

Reviewer #1:

Remarks to the Author:

The authors has carefully responded to this reviewer's comments and has addressed all of the reviewers concerns and suggestions. I support publication of this manuscript.

Reviewer #2:

Remarks to the Author:

The authors have answered the reviewer's comment. The reviewer believe the manuscript is suitable for publication in Nature Communications.

Reviewer #3:

Remarks to the Author:

My comments were fully addressed. Of course, this manuscript was recommended for publication in Nature Communication.

REVIEWERS' COMMENTS

Reviewer #1:

The authors has carefully responded to this reviewer's comments and has addressed all of the reviewers concerns and suggestions. I support publication of this manuscript.

Reviewer #2:

The authors have answered the reviewer's comment. The reviewer believe the manuscript is suitable for publication in Nature Communications.

Reviewer #3:

My comments were fully addressed. Of course, this manuscript was recommended for publication in Nature Communication.

Response: We thank all the three reviewers for supporting publication of our work in Nature Communication.